



# Magnetic field fluctuation properties of coronal mass ejection-driven sheath regions in the near-Earth solar wind

Emilia K. J. Kilpua[1], Dominique Fontaine[2], Simon Good[1], Matti Ala-Lahti[1], Adnane Osmane[1], Erika Palmerio[1,3], Emiliya Yordanova[4], Clement Moissard[2], Lina Z. Hadid[5,4], and Miho Janvier[6]

[1]Department of Physics, University of Helsinki, Helsinki, Finland
[2]LPP, CNRS, Ecole Polytechnique, Sorbonne Université, Université Paris Saclay, Observatoire de Paris, Institut Polytechnique de Paris, PSL Research University, Palaiseau, France
[3]Space Sciences Laboratory, University of California–Berkeley, Berkeley, CA, USA
[4]Swedish Institute of Space Physics, Uppsala, Sweden
[5]ESTEC, European Space Agency, Noordwijk, Netherlands
[6]Université Paris-Saclay, CNRS, Institut d'astrophysique spatiale, 91405, Orsay, France

*Correspondence to:* Emilia Kilpua (emilia.kilpua@helsinki.fi)

**Abstract.** In this work, we investigate the magnetic field fluctuations in three coronal mass ejection (CME)-driven sheath regions at 1 AU with their speeds ranging from slow to fast. The data set we use consists primarily of high resolution (0.092 s) magnetic field measurements from the Wind spacecraft. We analyse magnetic field fluctuation amplitudes and fluctuation amplitudes normalised to the mean magnetic field, compressibility, and spectral properties of fluctuations. We also analyse
intermittency using various approaches: we apply the partial variance of increments (PVI) method, investigate probability distribution functions of fluctuations, including their skewness and kurtosis, and perform a structure function analysis. Our analysis is conducted separately for three different subregions in the sheath and in the solar wind ahead of it, each 1 hr in duration. We find that, for all cases, the transition from the solar wind ahead to the sheath generates new fluctuations and the intermittency and compressibility increase, while the region closest to the ejecta leading edge resembled the solar wind ahead.
The spectral indices exhibit large variability in different parts of the sheath, but are typically steeper than Kolmogorov's in the inertial range. The structure function analysis produced generally much better fit with the extended $p$-model (Kraichnan's form) than with the standard version, implying that turbulence is not fully developed in CME sheaths near Earth's orbit. The $p$-values obtained ($p \sim 0.8$–$0.9$) also suggest relatively high intermittency. At the smallest timescales investigated, the spectral indices indicate relatively shallow slopes (between $-2$ and $-2.5$), suggesting that in CME-driven sheaths at 1 AU the
energy cascade from larger to smaller scales could still be ongoing through the ion scale. Regarding many properties (e.g., spectral indices and compressibility) turbulent properties in sheaths, regardless their speed, resemble that of the slow wind, rather fast wind. They are also partly similar to properties reported in terrestrial magnetosheath, in particular regarding their intermittency, compressibility and absence of Kolmogorov's type turbulence. Our study also reveals that turbulent properties can vary considerably within the sheath. This was in particular the case for the fast sheath behind the strong and quasi-parallel
shock, including a small, coherent structure embedded close to its midpoint. Our results support the view of the complex formation of the sheath and different physical mechanisms playing a role in generating fluctuations in them.





**Keywords.** Coronal mass ejections, Space plasma physics, Solar wind

## 1 Introduction

Coronal mass ejection (CME)-driven sheath regions (e.g., Kilpua et al., 2017a) are turbulent large-scale heliospheric structures that are important drivers of disturbances in the near-Earth environment and present a useful natural laboratory to study many
fundamental plasma physical phenomena. Sheaths form gradually as the CME travels through the solar wind from the Sun to Earth and they present properties of both expansion and propagation sheaths (e.g., Siscoe and Odstrcil, 2008). As sheaths accumulate from layers of inhomogeneous plasma and magnetic field ahead, discontinuities and reconnection exhausts can be commonly found (e.g., Feng and Wang, 2013). Magnetic field fluctuations in the sheaths are both transmitted from the preceding solar wind and generated within the sheath, e.g., via physical processes at the shock and due to draping of the
magnetic field around the driving ejecta (e.g., Gosling and McComas, 1987; Kataoka et al., 2005; Siscoe et al., 2007). Sheaths also embed various plasma waves; for instance, mirror mode and Alfvén ion cyclotron waves are frequently found (Ala-Lahti et al., 2018, 2019). The compressed and turbulent nature of the sheaths enhances solar wind–magnetosphere coupling efficiency when they interact with Earth's magnetic environment (Kilpua et al., 2017b). Sheaths have indeed been shown to be important drivers of geomagnetic storms (e.g., Tsurutani et al., 1988; Huttunen et al., 2002; Zhang et al., 2007; Echer et al., 2008;
Yermolaev et al., 2010); in particular, they often cause intense responses in the high-latitude magnetosphere and ionosphere (e.g., Huttunen and Koskinen, 2004; Nikolaeva et al., 2011), they are related to intense and diverse wave activity in the inner magnetosphere (e.g., Kalliokoski et al., 2019; Kilpua et al., 2019b), and produce drastic variations in high-energy electron fluxes in the Van Allen radiation belts (e.g., Hietala et al., 2014; Kilpua et al., 2015; Lugaz et al., 2015; Alves et al., 2016; Turner et al., 2019).

Despite their importance for heliophysics and solar–terrestrial studies, CME-driven sheath regions are currently relatively little investigated and their multi-scale structure is not well understood. In particular, a better understanding of the nature and properties of the magnetic field fluctuations in sheaths and how these properties vary spatially within them are crucial for understanding their formation and space weather impact. Detailed studies of magnetic field fluctuations in sheaths are also expected to yield new insight into outstanding problems in turbulence research, such as how intermittency in turbulence is
related to the formation of coherent structures and discontinuities (e.g., current sheets) in plasmas.

    Good et al. (2020) investigated the radial evolution of magnetic field fluctuations for a CME-driven sheath using almost radially aligned observations made by MESSENGER at ~0.5 AU and STEREO-B at ~1 AU. At ~0.5 AU, where the leading-edge shock was quasi-parallel, the downstream sheath plasma developed a range of large-angle field rotations, discontinuities and complex structure that were absent in the upstream wind, with a corresponding steepening of the inertial-range spectral
slope and increase in mean fluctuation compressibility in the sheath. At ~1 AU, in contrast, the shock crossing was quasi-perpendicular and much less difference in fluctuation properties between the sheath plasma and upstream wind was observed. Intermittency in the sheath turbulence grew between the spacecraft. The shock–sheath transition at MESSENGER had a qualitatively similar 'ageing' effect on the plasma to that seen in the ambient solar wind with radial propagation between the





two spacecraft. Moissard et al. (2019) performed a statistical investigation of magnetic field fluctuations in 42 CME-driven sheath regions observed in the near-Earth solar wind. In particular, the authors studied compressibility $C = P_{||}/(P_\perp + P_{||})$ and anisotropy $A = P_\perp/(2P_{||})$, where $P_{||}$ and $P_\perp$ are the parallel and perpendicular power of fluctuations, respectively. They found that sheaths present increased compressibility and lower anisotropy when compared to the preceding solar wind or to

the following CME ejecta. The total fluctuation power was also considerably ($\sim$10 times) higher in the sheath than in the surrounding solar wind, consistent with some earlier studies (e.g., Kilpua et al., 2013). The fluctuation power was also found to be predominantly in the direction perpendicular to the magnetic field.

Properties of the driving CME and preceding solar wind are likely to have a significant role for turbulence in sheaths. Kilpua et al. (2013) and Kilpua et al. (2019a) reported an increase in magnetic field fluctuation power with increasing CME speed.

Moissard et al. (2019) found this same tendency and also emphasised the importance of the magnetic field fluctuation power in the preceding solar wind. Riazantseva et al. (2019) analysed high-resolution SPEKTR-R spacecraft magnetic field data for fast and slow solar wind, magnetic clouds and non-cloud ejecta, sheaths and fast–slow stream interaction regions (SIRs; e.g., Richardson, 2018). They found that the inertial (magnetohydrodynamic) range spectral slopes in slow solar wind, in sheaths ahead of ejecta, and within magnetic clouds were less steep than the $f^{-5/3}$ power law for Kolmogorov turbulence, while the

slopes in fast solar wind, in sheaths ahead of magnetic clouds, and within ejecta were closer to the Kolmogorov index. In the kinetic range, in turn, fluctuation spectra at kinetic scales were steepest in compressive heliospheric structures, i.e. in sheaths (for both magnetic cloud and ejecta) and SIRs.

In this work, we analyse three CME-driven sheath regions in the near-Earth solar wind that present high, intermediate, and low speeds. The Mach numbers of the shocks preceding these sheaths range correspondingly from high (4.9) to low (1.8)

and their shock angles from quasi-parallel (33°) to almost perpendicular (86°). We investigate and compare distributions of the embedded magnetic field fluctuations, fluctuation amplitudes normalised to the mean magnetic field, compressibility, and intermittency. Our study is conducted for three separate 1-hour periods within the sheaths: near the shock, in the middle of the sheath, and close to the ejecta leading edge. This division is applied because properties in sheath regions often change considerably from the shock to the ejecta leading edge and magnetic field fluctuations in different parts can partly arise from

different physical processes (e.g., Kilpua et al., 2019b). We also include in the study one hour of the preceding solar wind for each event and investigate how fluctuation properties change from the solar wind to the sheath.

The manuscript is organised as follows: In Section 2 we describe the data sets used and some underlying assumptions. In Section 3 we present our analysis. First, we give a general overview of the three sheath events under study. Then, we investigate the properties of magnetic field fluctuations, spectral indices, and compressibility. Finally, we explore the intermittency of

fluctuations in more detail by analysing distribution functions, skewness, kurtosis, and structure functions. In Section 4 we discuss and in Section 5 we summarise our results.



## 2 Research data and assumptions

We primarily use high-resolution magnetic field observations from the Magnetic Fields Investigation (MFI; Lepping et al., 1995) instrument on board the Wind (Ogilvie and Desch, 1997) spacecraft. The magnetic field data during the observed events are available at 0.092-s cadence. We obtained the data through the NASA Goddard Space Flight Center Coordinated Data Analysis Web[1] (CDAWeb). To provide an overview of the wider solar wind conditions for the studied events, we also used plasma data from Wind's Solar Wind Experiment (SWE; Ogilvie et al., 1995) instrument, which are available at 90-s resolution. During the times of the events that are part of this study, Wind was located at the Lagrange L1 point.

In our analysis, we divide sheaths into three separate regions, each being one hour in duration. Furthermore, we consider a one-hour region of solar wind preceding the shock and sheath, excluding the 30-min period immediately ahead of the shock. The three distinct regions within the sheath are termed the Near-Shock, Mid-Sheath, and Near-Leading Edge (Near-LE) regions. The Near-Shock region extends from the shock to the sheath, excluding the 15 minutes immediately following the shock, and the Near-LE region ends 15 minutes before the leading edge of the ICME ejecta. The 15-min interval closest to the shock and the ejecta leading edge have been excluded to avoid the shock transitions and most immediate shock processes and uncertainties in the timing of the ejecta leading edge, respectively. The Mid-Sheath regions are located around the middle of the sheath, except for the fast event for which the Mid-Sheath region was selected to capture a coherent magnetic field structure embedded in the sheath. We however emphasise that the results for the Mid-Sheath are likely to depend strongly on the selected interval.

The fluctuations in the magnetic field are defined here as $\delta \boldsymbol{B} = \boldsymbol{B}(t) - \boldsymbol{B}(t + \Delta t)$, and fluctuation amplitude $\delta B = |\delta \boldsymbol{B}|$, where $\Delta t$ is the timescale or time-lag between two samples. We use 14 values of $\Delta t$ that range from 0.092 to 736 s (12.3 min), where the values are successively doubled. The observations thus cover most of the inertial range ($10^1$ s $\lesssim \Delta t \lesssim 10^3$ s) and the upper part of the sub-ion (kinetic) range ($\Delta t \lesssim 10^1$ s).

We study magnetic field fluctuations in the spacecraft frame, which is in relative motion with respect to the solar wind frame. To justify the transformation from spacecraft frequency to wavenumber, and to relate observed timescales to length scales in the plasma, the so-called Taylor hypothesis (Taylor, 1938; Matthaeus and Goldstein, 1982) must be valid. The hypothesis states in this context that when the timescales of the magnetic field fluctuations are much less than the timescale of rapidly flowing solar wind, the path of a spacecraft travelling through solar wind represents an instantaneous spatial cut. This can be expressed in a simple way as $\kappa = v_A/v \lesssim 1$ (Howes et al., 2014), where $v_A$ is the Alfvén speed, and $v$ the solar wind speed. The results are shown in Table 1, where it can be seen that for all events and subregions the criterion is met in this study.

## 3 Analysis and results

### 3.1 Event overview

We analyse three CME-driven sheath regions detected by the Wind spacecraft in the near-Earth solar wind at the Lagrange L1 point. The key parameters of these events are listed in Table 1. The events were selected from the list of 81 sheaths published

---

[1]http://cdaweb.gsfc.nasa.gov/





**Table 1.** Summary of the events analysed. The first two rows give the shock and CME ejecta leading edge (LE) times at Wind. The following rows give the shock parameters: magnetosonic Mach number ($M_{ms}$), shock angle ($\theta_{Bn}$, i.e., the angle between the shock normal and upstream magnetic field), shock speed ($V_{shock}$), and upstream plasma beta ($\beta_u$). The next rows give the sheath parameters: the duration of the sheath ($\Delta T$), the average solar wind speed in the sheath ($\langle V_{sheath}\rangle$), the average magnetic field magnitude ($\langle B_{sheath}\rangle$), the parameter $\kappa$, defined as $\kappa = v_A/v$ and that tests the validity of the Taylor hypothesis (see Section 2 for details), and the 'PMS coverage', which indicates the percentage of the sheath sub-regions occupied by planar magnetic structure (PMS)-type field behaviour (see Section 3.1). The last rows give the average solar wind speed and $\kappa$ values for the one-hour solar wind interval preceding the shock.

|  | Dec 14, 2006 (Fast) | Oct 24, 2001 (Medium speed) | Oct 31, 2012 (Slow) |
|---|---|---|---|
| Time |  |  |  |
| Shock [UT] | 12/14/2006 13:51 | 10/24/2011 17:39 | 10/31/2012 14:28 |
| Ejecta LE [UT] | 12/14/2006 22:36 | 10/25/2011 00:21 | 10/31/2012 23:30 |
| Shock |  |  |  |
| $M_{ms}$ | 4.9 | 2.5 | 1.8 |
| $\theta_{Bn}$ [°] | 33 | 64 | 86 |
| $V_{shock}$ [km/s] | 919 | 542 | 391 |
| $\beta_u$ | 5.2 | 1.4 | 5.2 |
| Sheath |  |  |  |
| $\Delta T$ [hours] | 8.7 | 6.9 | 9.0 |
| $\langle V_{sheath}\rangle$ [km/s] | 880 | 503 | 351 |
| $\langle B_{sheath}\rangle$ [nT] | 11.5 | 15.5 | 9.4 |
| $\kappa$ | 0.11–0.11–0.073 | 0.15–0.11–0.16 | 0.095–0.13–0.14 |
| PMS coverage | 0%–100%–100% | 100%–0%–0% | 0%–100%–100% |
| Preceding solar wind |  |  |  |
| $\langle V_{sw}\rangle$ [km/s] | 584 | 340 | 290 |
| $\kappa$ | 0.12 | 0.13 | 0.12 |





in Kilpua et al. (2019b) to have their speeds to represent the lowest, median and highest speeds of the whole population. The speed of the sheath was selected as our primary parameter for the event selection since solar wind turbulence studies often consider slow and fast wind separately due to their different evolution and origin, which affects their turbulent properties. We also required that the selected sheaths had at least 4 hours in duration, were followed by a well-defined CME ejecta that

was classified as a magnetic cloud in the Richardson and Cane ICME list[2] (Richardson and Cane, 2010), presented a clear transition from the sheath to the ejecta, and that no other CMEs were present within a one-day period prior to the interplanetary shock ahead of the sheath. The selected events are: December 14–15, 2006 (fast sheath), October 24–25, 2011 (medium-speed sheath), and October 31, 2012 (slow sheath) with their mean speeds calculated over the duration of the whole sheath being 880, 503, and 351 km/s, respectively. Table 1 also shows that the investigated events have also different shock strengths and

shock speeds. The fast sheath was preceded by the strongest and fastest shock (4.9 and 919 km/s), while the slow event was preceded by the weakest and slowest shock (1.8 and 391 km/s). The shock angles ($\theta_{Bn}$, i.e., the angle between the shock normal and upstream magnetic field) were also varied amongst the events; while the fastest sheath was associated with a quasi-parallel shock (33°), the intermediate and slow sheaths were preceded by quasi-perpendicular shocks. For the slowest sheath the shock angle was very close the perpendicular ($\theta_{Bn} = 86°$) The shock configuration is also expected to affect magnetic field

fluctuation properties in the sheath, in particular in the near-shock region (e.g., Bale et al., 2005a; Burgess et al., 2005).

Figure 1 shows the solar wind conditions during the three events under study. The orange-shaded intervals show the one-hour regions subject to a more detailed analysis. The first four panels give the interplanetary magnetic field (IMF) magnitude, IMF components in GSE coordinates (blue: $B_X$, green: $B_Y$, red: $B_Z$), solar wind speed, and density. It is clear that all three cases present well-defined sheath regions with large-amplitude magnetic field fluctuations embedded, and enhanced solar wind

density. It is however evident that the overall properties of the sheath vary considerable between the three events and from the shock to ejecta leading edge. We also note that the fast sheath is preceded by fast solar wind, while the medium-speed and slow sheaths are preceded by slow wind.

The last panel of Figure 1 gives the normalised partial variance of increments (PVI). The PVI parameter is defined as

$$\mathrm{PVI} = \frac{|\delta \boldsymbol{B}|}{\sqrt{\langle |\delta \boldsymbol{B}|^2 \rangle}}, \tag{1}$$

where the average in the denominator is taken over the whole interval shown. This quantity is used to detect coherent structures or discontinuities (e.g., Greco et al., 2018; Zhou et al., 2019). We calculate the PVI here using three different time-lags between two data points: $\Delta t = 0.18$, 24, and 186 s (0.18 s corresponding to the kinetic range, and 24 and 186 s corresponding to the inertial range). The figure shows that the PVI exhibits a number of spikes throughout most of the sheaths investigated, suggesting that intermittent structures are frequently present. The largest PVI spikes in the fast and medium-speed sheath

occur close to the shock, while for the slow sheath they are found close to the middle part of the sheath. The coherent structure in fast sheath exhibits low PVI. Another interesting feature visible from the PVI panel is that the Near-LE regions, in particular for the medium-speed and slow sheaths, have similar PVI levels as the solar wind ahead of them (excluding the peaks at the sheath-ejecta transition).

---

[2]http://www.srl.caltech.edu/ACE/ASC/DATA/level3/icmetable2.htm





Table 1 shows the percentage of the sheath sub-regions occupied by planar magnetic structures (PMSs) determined with the method described in Palmerio et al. (2016). PMSs are periods during which the variations of the magnetic field vectors remain nearly parallel to a fixed plane over an extended time period (e.g., Nakagawa et al., 1989; Jones and Balogh, 2000). PMSs typically cover a significant part of the sheath and can be present in all parts of the sheath (Palmerio et al., 2016). In

CME-driven sheaths, they are thought to arise from processes at the shock (i.e. the alignment and amplification of pre-existing discontinuities; e.g. Neugebauer et al., 1993; Kataoka et al., 2005) and field-line draping around the CME ejecta (e.g., Gosling and McComas, 1987). As reported by Kataoka et al. (2005) and Palmerio et al. (2016), PMSs are most frequently found behind strong quasi-perpendicular shocks with high upstream plasma beta. The fast sheath and the slow one had no PMS in their Near-Shock region, but their Mid-Sheath and Near-LE regions were fully covered by planar fields. The lack of PMSs in the

Near-Shock region for the fast sheath is likely due to the quasi-parallel shock configuration, while for the slow sheath it is probably due to its leading shock being weak ($M_{ms} = 1.8$). The medium-speed sheath presented in turn planar fields in the Near-Shock region, but no PMS in the other subregions. The comparison of planar periods with the PVI values shown in Figure 1 does not reveal an obvious correlation.

### 3.2   Distributions and averages of magnetic field fluctuations

Figure 2 shows the Probability Distribution Functions (PDFs) of the normalised fluctuation amplitude $\delta B/B$ for timescales $\Delta t$ ranging from 0.092 s (light red) to 736 s (dark red). The columns show from left to right different subregions and from top to bottom the three different events.

The solar wind preceding the medium-speed and slow sheaths have practically all $\delta B/B$ values $< 1$ (left panels of Figure 2), meaning that there are no significant rotations of the field direction (no larger than $60°$ for pure rotation case). The fast event

(Dec 14, 2006), in turn, is preceded by a solar wind with clearly larger normalised fluctuation amplitudes. The $\delta B/B$ values are nevertheless mostly $< 2$, consistent with the absence of strongly compressive fluctuations. Note that for purely Alfvénic (i.e. incompressible) fluctuations, for which the field magnitude does not change, $\delta B/B$ must be $< 2$. We also note that the solar wind ahead of the fast sheath could be affected by foreshock waves, since it is ahead of the quasi-parallel shock as discussed in Section 3.1.

Comparison of the preceding solar wind and Near-Shock regions shows that the normalised fluctuation amplitudes are spread to considerably larger values in the Near-Shock region for all events and timescales investigated. This can be due to the generation of new fluctuations or amplification (in magnitude and/or change in the field direction) of pre-existing fluctuations relative to the mean field. For the medium-speed and slow sheath distributions, $\delta B/B$ values are mostly confined to $< \sqrt{2}$ in the Near-Shock region, which means that there are no rotations of the field exceeding $90°$, while for the fast sheath there is a

significant fraction of values $> 2$ for scales $\Delta t \gtrsim 6$ s, meaning that large rotations of the field direction exist and fluctuations must be at least partly compressional.

Figure 2 clearly shows that the PDFs vary considerably in different sheath subregions. This is particularly evident in the fast sheath: Both the Near-Shock and Near-LE regions have PDFs extending to $\delta B/B > 2$, but for the Near-LE region this occurs for the larger time-scales only ($\Delta t \gtrsim 48$ s), i.e., at scales within the inertial range, while for the Near-Shock regions PDFs





populate $\delta B/B > 2$ for $\Delta t \gtrsim 1.5$ s. Within the small coherent structure near the middle of the fast sheath, fluctuations are in turn restricted to $\delta B/B \leq 1$, thus being much smaller than in the other parts of the sheath or preceding solar wind. For the medium and slow-speed sheaths, the largest $\delta B/B$ values occur in the Mid-Sheath region and in the case of the medium-speed sheath $\delta B/B$ exceeds 2 for the largest scales.

One interesting feature in Figure 2 is that the Near-LE distributions resemble quite closely those in the solar wind ahead, both in their extent to higher $\delta B/B$ values and shape of the curves. This is most distinct for the medium-speed and slow sheaths, for which $\delta B/B$ values are mostly confined to $< 1$. For the fast sheath distributions in the Near-LE region at larger scales are however clearly flatter and less smooth than in the solar wind ahead (or in the Near-Shock region).

Figure 2 also reveals the expected general trend; for the smallest timescales, PDFs have sharp peaks at low $\delta B/B$ values
and then decay exponentially, while at larger timescales distributions become broader, in particular in the sheaths. This behaviour, i.e. that distributions of normalised fluctuations become more Gaussian and shift towards larger values with increasing timescale, is consistent with previous studies carried out in the solar wind (e.g., Sorriso-Valvo et al., 2001; Chen et al., 2015; Matteini et al., 2018).

We now consider the average fluctuation amplitudes across the range of timescales. The top and middle panels of Figure 3
show the mean values of $\delta B$ and $\delta B/B$ as a function of timescale. In the top panel, the ion cyclotron timescales are also plotted, using the mean magnetic field magnitude over the whole subregion in question.

In the preceding solar wind intervals, mean fluctuation amplitudes $\langle \delta B \rangle$ and normalised fluctuation amplitudes $\langle \delta B/B \rangle$ are highest ahead of the fast sheath and lowest ahead of the slow sheath. We however note that differences are partly relatively small, in particular for the mean fluctuation amplitudes between the fast and medium-speed sheath. As is to be expected
considering the field compression at the shock, $\langle \delta B \rangle$ are considerably higher in the Near-Shock region than in the preceding solar wind for all events and timescales. The $\langle \delta B/B \rangle$ are also higher in the Near-Shock and Mid-Sheath regions than in the solar wind ahead, also reported in Good et al. (2020). This suggests that the transition from the solar wind to the sheath generates new and/or enhances pre-existing magnetic field fluctuations at all timescales.

Matteini et al. (2018) reported that the spectra of normalized fluctuation amplitudes collapsed on the same curve for the
solar wind periods observed by Helios and Ulyssses suggesting the modulation of the field fluctuations with the magnetic field magnitude. Their results were however obtained at varying heliospheric distances and solar wind speeds. In our study $\langle \delta B \rangle$ curves are organised according to the solar wind speed as mentioned above. This trending is maintained throughout the sheath, except for the small coherent structure in the mid-point of the fast sheath that exhibits both lower $\langle \delta B \rangle$ and $\langle \delta B/B \rangle$ values (especially so at larger scales) than in the solar wind ahead or in the Mid-Sheath regions of the other two sheaths.

## 3.3   Spectral indices

Several studies have investigated how the power spectrum of solar wind magnetic field fluctuations is in agreement with predictions made by turbulence theories (e.g., Coleman, 1968; Bavassano et al., 1982; Horbury and Balogh, 2001; Bale et al., 2005b; Tsurutani et al., 2018; Verscharen et al., 2019). Kolmogorov's spatially homogeneous hydrodynamic turbulence model gives $f^{-5/3}$ (i.e., spectral index $\alpha = -1.67$; Kolmogorov, 1941) and is based on the assumption that energy cascades from larger





to smaller scales through eddies that break down evenly and are space filling. The modification of Kolmogorov's model for a magnetohydromagnetic fluid, based on works by Iroshnikov (1964) and Kraichnan (1965), takes into consideration interactions between oppositely propagating Alfvén waves and equipartitioning between magnetic and kinetic energy. In the inertial regime of the Kraichnan–Iroshinikov model, the energy spectrum is proportional to $f^{-3/2}$ (i.e., spectral index $\alpha = -1.5$), i.e. less

steep than in the Kolmogorov model. The energy is then dissipated in the kinetic range and the breakpoint occurs around the ion cyclotron scale ($t_{ci} = 1/f_{ci}$, where $f_{ci}$ is the ion cyclotron frequency).

    The kinetic regime exhibits a steeper spectral slope than the inertial regime, with spectral index $\alpha_k \simeq -2.8$ typically reported in the solar wind (e.g., Alexandrova et al., 2013; Bruno et al., 2017; Huang et al., 2017). The timescales studied here should generally be below the energy-driving $f^{-1}$ regime, which occurs at frequencies $f \gtrsim 10^{-3}$ Hz for the fast wind, i.e., timescales

$\gtrsim$16.7 min (e.g., Bruno and Carbone, 2013) and at frequencies $f \gtrsim 10^{-4}$ Hz, i.e., timescales $\gtrsim$166.7 minutes or 2.7 hours for the slow wind (e.g., Bruno, 2019). This regime known as the "energy containing scale" is likely composed of fluctuations with various origins and remains debated (see e.g. discussion in Bruno et al., 2019). We note that some of the curves in the top panels of Figure 3, in particular the preceding solar wind and Near-Shock subregions in the fast sheath, exhibit a flattening trend towards the largest timescales that could indicate the transition to the $f^{-1}$ regime. Other possibility is that, as discussed

in Section 3.2, foreshock waves can affect, and could result to a small hump at timescales around 100 s.

    In the top panels of Figure 3 the pink dashed lines $l^{0.33}$ show the Kolmogorov scaling and the cyan dash-dotted lines $l^{0.25}$ the Kraichnan–Iroshinikov scaling in the inertial range. Note that the $l^{0.33}$ and $l^{0.25}$ given in the figure correspond to $f^{-5/3}$ and $f^{-3/2}$, respectively (e.g., Matteini et al., 2018). In the kinetic range we have plotted the scaling $l^{0.9}$ ($\sim f^{-2.8}$). Table 2 shows the kinetic and inertial range spectral indices for the preceding solar wind and in the three subregions of the sheath.

The kinetic range indices are determined using the first three timescales from 0.092 to 0.37 s and the inertial range indices using three timescales from 24 to 96 s. The timescales used to calculate the inertial scale indices are above the ion cyclotron timescales for all cases and fall into the region where the $\langle \delta B \rangle$ curves have approximately a linear behaviour (see Figure 3). They are also well above the 0.3 Hz (3 s) of the dissipation range breakpoint found in the extensive statistical study at L1 by Smith et al. (2006).

In the kinetic range, the spectral slopes are consistently less steep than the $-2.8$ index cited above. For the solar wind preceding the slow sheath, the kinetic range spectral index is $-1.72$, while in other regions they are distributed between $-2.23$ and $-2.49$. In the inertial range the spectral indices vary considerably. The Near-Shock region for the fast sheath, Near-Shock and Near-LE regions for the medium-speed sheath, and the preceding solar wind region for the slow sheath have their inertial range spectral indices matching or close to the Kolmogorov index ($-1.67$). The Mid-Sheath region for the fast sheath and the

preceding solar wind for the medium-speed sheath exhibit in turn their spectral index close to the Kraichnan–Iroshinikov value ($-1.5$). Otherwise, the spectral indices are clearly steeper than Kolmogorov's. For the fast sheath, the slope in the preceding solar wind region could have been affected by the possible foreshock wave related hump (see above) and the slope could be in reality closer to Kolmogorov's.





**Table 2.** Spectral indices calculated by fitting a straight line in log space for the kinetic ($\Delta t = 0.092$–$0.37$ s) and inertial ($\Delta t = 24$–$96$ s) ranges. The third column gives the ion cyclotron period for each region. The last column gives $\kappa = v_A/(v|cos\theta|)$ that tests the validity of the Taylor hypothesis (see Section 2 for details).

|  | Kinetic | Inertial | $t_{ci}$ [sec] | $\kappa$ |
|---|---|---|---|---|
| **Fast** | | | | |
| Preceding SW | -2.47 | -1.84 | 1.28 | 0.16 |
| Near-Shock | -2.49 | -1.67 | 0.87 | 0.17 |
| Mid-Sheath | -2.23 | -1.49 | 0.53 | 0.21 |
| Near-LE | -2.27 | -1.99 | 0.28 | 0.18 |
| **Intermediate speed** | | | | |
| Preceding SW | -2.29 | -1.56 | 1.25 | 0.16 |
| Near-Shock | -2.41 | -1.71 | 0.62 | 0.28 |
| Mid-Sheath | -2.42 | -2.03 | 0.72 | 0.18 |
| Near-LE | -2.44 | -1.63 | 0.28 | 0.21 |
| **Slow** | | | | |
| Preceding SW | -1.72 | -1.67 | 0.93 | 0.12 |
| Near-Shock | -2.37 | -1.84 | 0.77 | 0.15 |
| Mid-Sheath | -2.56 | -1.81 | 0.75 | 0.17 |
| Near-LE | -2.15 | -1.77 | 0.80 | 0.28 |

## 3.4 Compressibility

The bottom panels of Figure 3 show the mean of $\delta|B|/\delta B$. This parameter gives the compressibility of magnetic fluctuations, i.e. the mean amount of compression as a fraction of the total fluctuation amplitude, as a function of timescale. The horizontal line is at $\delta|B|/\delta B = 0.2$ (Matteini et al., 2018), which represents the typical upper threshold of the level of compressibility

5    in magnetic field fluctuations found for the fast solar wind in the inertial regime; the slow solar wind tends to have $\delta|B|/\delta B$ values exceeding $0.2$.

For all cases investigated, compressibility increases from the solar wind ahead to the Near-Shock region, suggesting that the locally-generated new fluctuations in the sheath are at least partly compressible. This occurs for all timescales for the fast and slow sheaths. For the medium-speed sheath the two largest scales show a decrease, with compressibility values $< 0.2$. Again,

10    it is possible that this decrease at the largest scales is due to a larger statistical error, since the magnetic field magnitude does not change that much in this subregion when compared to other events. The solar wind preceding the fast sheath has $\delta|B|/\delta B$ values below $0.2$ in the inertial range, consistent with its speed being in the fast-wind range (see Figure 1).

Figure   3 shows that the level of compressibility varies considerably in different subregions of the sheath. For the fast sheath, compressibility is relatively high in all sheath subregions and above the preceding solar wind values, in particular in the





small coherent structure near the middle of the sheath. This is in agreement with normalized fluctuation amplitudes $\langle \delta |B|/B \rangle$ being low in the region (Section 3.2), as compressible/parallel fluctuations are known to have lower amplitudes/power than Alfvénic/perpendicular fluctuations (i.e., there is power anisotropy). For the medium speed sheath, compressibility values are also high in the Mid-Sheath, but in the Near-LE region in turn, fluctuations are even less compressible than in the preceding

solar wind and for the most time-scales below 0.2. The slow sheath has most compressible fluctuations in the Near-Shock and Near-LE regions, comparable to values for the fast sheath, but in the Mid-Sheath region compressibility values fall below the preceding solar wind values and even the 0.2 threshold for the largest timescales ($\Delta t > 96$ s). In all instances, it can be seen that compressibility generally decreases from the smallest scales until about 100 s. This is a well known trend from previous solar wind studies (e.g., Chen et al., 2015; Matteini et al., 2018; Good et al., 2020). We note that the increase for the few highest

timescales could be related to statistical errors in the data as discussed above.

### 3.5 Intermittency

Several studies have shown that fluctuations in the solar wind are typically strongly intermittent (e.g., Burlaga, 1991; Feynman and Ruzmaikin, 1994; Marsch and Tu, 1994, 1997; Pagel and Balogh, 2001; Yordanova et al., 2009). Intermittency describes inhomogeneity in the energy transfer between scales, and is manifested as a lack of self-similarity in fluctuation distributions

between scales (see, e.g., reviews by Horbury et al., 2005; Sorriso-Valvo et al., 2005; Bruno, 2019; Verscharen et al., 2019). In the solar wind, it can arise from coherent structures such as current sheets and discontinuities. The PVI parameter shown in the bottom panel of Figure 1 already gives some indication that intermittent structures were embedded throughout the sheaths analysed.

#### 3.5.1 Probability distribution functions

Deviations from a Gaussian distribution in fluctuation PDFs at different timescales can reveal the presence of intermittency. Figure 4 shows distributions of $(\delta B_Z - \mu)/\sigma$ for different $\Delta t$ values. Here, $\sigma$ is the standard deviation and $\mu$ the mean of $\delta B_Z$ calculated using a 15-minute sliding average. We investigate here the $Z$-component of the IMF since it has the largest importance for the solar wind magnetosphere coupling and geoefficiency of the sheath. The black dashed line shows the normal distribution whose standard deviation and mean correspond that of the $(\delta B_Z - \mu)/\sigma$ distribution calculated using the 6-second

timescale, but we note that normal distributions for the other shown timescales would be almost inseparable from the 6-second curve. The distributions are thus clearly non-Gaussian, and particularly so at smaller timescales (light red curves) where they have flatter tails: these are clear signs of intermittency. At larger timescales (dark red curves), distributions become generally more Gaussian, consistent with, e.g., Greco et al. (2008). Non-self-similarity is also qualitatively evident in Figure 4.

#### 3.5.2 Skewness and kurtosis

In order to characterise the non-Gaussian aspects of the distributions, we compute the higher-order moments. Figure 5 gives the skewness and kurtosis calculated for the distributions shown in Figure 4. Skewness is related to the third distribution





moment and gives information on the degree of distribution asymmetry. For the normal distribution, skewness is zero, i.e., the distribution is symmetric around the mean value. Positive skewness indicates an extended tail at larger values than the mean (i.e., weighting and a longer tail and towards the right), whereas negative skewness indicates an extended tail at smaller values than the mean (i.e., weighting and a longer tail towards the left). Kurtosis is related to the fourth distribution moment and can

be used as the proxy of large fluctuations that are indication of intermittency (e.g., Krommes, 2002; Osmane et al., 2015). PDFs with long tails have larger kurtosis than narrow PDFs. For the normal distribution, kurtosis is 3; values larger than 3 indicate flatter tails while values less than 3 indicate lighter tails. In Figure 5 we have subtracted the value 3 from the kurtosis, such that 0 depicts the normal distribution for both kurtosis and skewness.

For all investigated subregions skewness has mostly small absolute values ($< 1$), suggesting that distributions and fluctua-

tions are generally symmetric around the mean ($\sim 0$ nT). There are no obvious drastic differences in skewnesses between the preceding solar wind and the sheath. One feature visible from the plot is that for all events the skewnesses in the preceding solar wind tend to have negative values, signifying that there is an excess of negative fluctuations, while in the sheath both negative and positive skewness values are observed more evenly. The largest absolute magnitudes of skewnesses (i.e., indicating largest asymmetries when compared to Gaussian distribution) are found for the medium-speed and slow sheaths in the Mid-Sheath

region.

The kurtosis values are nearly all positive, indicating an excess of high-amplitude $\delta B_Z$ fluctuations with respect to the normal distribution. In the majority of cases, kurtosis decreases towards zero through the inertial range, indicating that distributions become more Gaussian with increasing timescale. This is consistent with the qualitative assessment of distribution shapes discussed in Section 3.5.1. In the solar wind, kurtosis peaks broadly near the small-scale end of the inertial range, and reduces

in value with decreasing scale through the kinetic range (most clear for the medium-speed and slow event). Across the same scales in the sheaths, in contrast, kurtosis flattens or continues to increase with reducing scale. Peaks in kurtosis are associated with the inertial–kinetic spectral breakpoint (e.g., Chen et al., 2015); the ion cyclotron period (Table 1), which is in the vicinity of this spectral break, is larger in the preceding solar wind than in the sheaths, which may partly explain the more obvious kurtosis peaks. There is also a clear general increase in kurtosis between the solar wind ahead and Near-LE regions for the

slow and intermediate speed events, while there is no significant change in values for the fast event. This could indicate that the Near-LE distributions are less peaked for the fast sheath, i.e. in contrast to the behaviour of the medium-speed and slow sheaths.

### 3.5.3 Structure function analysis

Intermittency can be also investigated using structure functions (e.g., Bruno and Carbone, 2013). The structure function of

order $m$ of the fluctuation amplitude in the $i$th magnetic field component is

$$S_B^m(\Delta t) = \langle |\delta B_i| \rangle = \langle |B_i(t) - B_i(\Delta t + t)|^m \rangle. \tag{2}$$

If the system conforms to a self-similar scaling law, then $S_B^m(\Delta t) \sim \Delta t^{g(m)}$. The scaling exponent is $g(m) = m/3$ (i.e., $S_B^m(\Delta t) \sim \Delta t^{m/3}$) in Kolmogorov's turbulence and $g(m) = m/4$ in the Kraichnan–Iroshinikov model.



To account for intermittency and inhomogenieties in the energy transfer between scales, various turbulence models have been proposed in the literature (see a more detailed description from e.g. Horbury and Balogh, 1997; Bruno, 2019). Several models have been shown to fit observations reasonably well; for brevity, only results for the $p$-model are shown here. We note that other models were tested, e.g. the random $\beta$-model (Frisch et al., 1978; Paladin and Vulpiani, 1987) and the She–Levyque

model (She and Leveque, 1994), but it was found that the $p$-model agreed best with the observed structure function scaling.

The standard $p$-model by Meneveau and Sreenivasan (1987) (see also Tu et al., 1996) assumes fully-developed turbulence in which intermittency arises from the unequal breakdown of eddies, i.e., resulting daughter eddies have different amount of energy. The structure function scaling exponents in the $p$-model are defined as

$$g(m) = 1 - log_2[p^{m/q} + (1-p)^{m/q}], \tag{3}$$

where the intermittency parameter $p$ varies between $0.5$ and $1$. The parameter $q$ is $3$ in the Kolmogorov form and $4$ for the Kraichnan–Iroshinikov form (e.g., Carbone, 1993). The $p$ value of $0.5$ corresponds to the non-intermittent case, giving $g(m) = m/3$ for Kolmogorov and $g(m) = m/4$ for Kraichnan–Iroshinikov turbulence. In the non-intermittent case, the structure function scaling exponents are linear with the moment $m$. Where there is intermittency, the scaling exponents flatten for larger $m$ and curves become nonlinear. The maximum intermittency in both cases occurs for $p = 1$, i.e. $g(m) = 1$.

The extended $p$-model(Tu et al., 1996; Marsch and Tu, 1997) describes turbulence that is not fully developed, as has been identified, for example, in the inner heliosphere. In the Kolmogorov form,

$$g(m) = (-\frac{5}{3} + \frac{3}{2}\alpha')\frac{m}{3} + 1 - log_2[p^{m/3} + (1-p)^{m/3}], \tag{4}$$

and in the Kraichnan–Iroshinikov form,

$$g(m) = (-3 + 2\alpha')\frac{m}{4} + 1 - log_2[p^{m/4} + (1-p)^{m/4}]. \tag{5}$$

Here two parameters, $p$ and $\alpha'$, are needed to describe the turbulence cascade, because the cascade depends on scale for under-developed turbulence. Thus the power spectral index is not linearly related to the second-order structure function exponent (as is the case for fully developed turbulence), but includes spatial inhomogeneity. The intermittency parameter, $p$, again gives the spatial inhomogeneity of the cascade rate, while the $\alpha'$ parameter is the intrinsic spectral slope describing the scaling properties of the space-averaged cascade rate. The first terms on the right-hand sides of the previous two equations now represent scale

dependence of the cascade, and the second terms represent intermittency. The spectral index is related to the intrinsic spectral index and $p$ parameter in the Kolmogorov's form

$$\alpha = \alpha' + \frac{1}{3} - log_2(p^{2/3} + (1-p)^{2/3}) \tag{6}$$

and in the Kraichnan–Iroshinikov form

$$\alpha = \alpha' + \frac{1}{2} - log_2(p^{2/3} + (1-p)^{1/2}) \tag{7}$$

Structure functions up to $m = 4$ only have been calculated since our 1-hour data intervals (with sampling resolution of $0.092$ s) include only approximately $39,130$ data points (see e.g. discussion in Horbury and Balogh, 1997). The scaling indices





are calculated over timescales from 24 to 94 s, within the inertial range (see Section 3.3). The results are only shown for $\delta B_z$, as discussed in the Introduction we here focus in this component.

In Figure 6, the standard $p$-model scaling exponent curves are plotted as a function of $m$ for $p = 0.5$–1. The observational results are shown as lime-green crosses (solar wind ahead) and dots (sheath subregions). Orange curves correspond to Kol-
mogorov (K) scalings and blue curves correspond to Kraichnan–Iroshinikov (K–I) scalings for varying amounts of intermittency, $p$. The grey horizontal line gives the maximum-intermittency case (i.e., $p = 1; g(m) = 1$), and the pink and cyan curves the non-intermittent (i.e., $p = 0.5$) cases corresponding to Kolmogorov and Kraichnan–Iroshinikov turbulence, respectively. The thick black curves show the best non-linear least squares fits to the extended $p$-model. We note that, for the majority of cases, the standard $p$-model curves do not match well with the observed $g(m)$-curve (lime dots and crosses); there is either
significant deviations between the observed and modelled points, or the observed points do not follow the model trends, or both. The best agreement with the standard $p$-model is with the solar wind ahead of the fast sheath, being consistent with the Kolmogorov form at high $p$-values ($\sim$0.9). In both intermittent and non-intermittent cases, the structure function should have either $g(3) = 1$ for the Kolmogorov scaling or $g(4) = 1$ for the Kraichnan–Iroshinikov scaling. There are several intervals in our data set for which neither $g(3) \approx 1$ nor $g(4) \approx 1$ hold. The observed values deviate from the standard $p$-model curves in
particular for the fast sheath subregions, the solar wind preceding the slow sheath and the Near-LE region for the slow sheath.

The extended $p$-model can be fitted with the data with excellent agreement and the Kolmogorov and Kraichnan–Iroshinikov forms overlap (black thick curve). However, these two forms of the model yield very different $p$ and $\alpha$ values; these values are given in the panels of Figure 6, with the corresponding standard deviation errors in parenthesis. The errors are of the order $10^{-3}$. We note that the Kraichnan–Iroshinikov form fits yield consistently larger $p$-values than the Kolmogorov form fits, but
both indicate high intermittency. The values of spectral indices given by the Kraichnan–Iroshinikov form are however clearly more consistent with the spectral indices shown in Table 2 that were calculate for the fitting for $\delta B$ using the same timescale range.

## 4  Discussion

We have investigated magnetic field fluctuations in three CME-driven sheath regions observed in the solar wind at the Lagrange
L1 point. Three parts of the sheath were studied separately, corresponding to the regions just adjacent to the shock (Near-Shock), at or near the middle of the sheath (Mid-Sheath), and adjacent to the ejecta leading edge (Near-LE). The results were also compared with the solar wind preceding the sheaths.

For all three cases, we found that the spectral and turbulent properties were considerably different between the preceding solar wind and the sheath. This was the case despite very different overall event properties, e.g., speed of the sheath and
preceding shock, shock angle and strength, level of upstream fluctuations and upstream solar wind speed and plasma beta. This is in contrast with the sheath analysed by Good et al. (2020), who found clear differences between the solar wind ahead and in the sheath at the orbit of Mercury, but not near the orbit of Earth. However, they investigated fluctuations throughout the sheath collectively, while we have investigated three sub-regions separately. The sheath analysed in their work was slow





and, at Earth's orbit, the preceding shock had a quasi-perpendicular configuration. We emphasised that in our study some clear changes between the solar wind ahead and the Near-Shock subregion in particular occurred also for the slowest sheath that was preceded by an almost-perpendicular shock.

Firstly, distributions of normalised magnetic field fluctuations were flatter and spread to considerably higher values in the
Near-Shock region than in the preceding solar wind for all events. For the fast sheath in particular $\delta B/B > 2$ values demonstrated the existence of significant rotations and compressional fluctuations. This is expected as the processes at the shock are more efficient the faster and stronger the shock is, e.g., fast shocks are expected to cause more efficiently alignment and amplification of pre-existing solar wind discontinuities, provide more free-energy for wave generation, etc. (e.g., Neugebauer et al., 1993; Kataoka et al., 2005; Kilpua et al., 2013; Ala-Lahti et al., 2018, 2019). The quasi-parallel nature of the shock
could have also enhanced the fluctuations for the fast event. For all cases, the distribution of the magnetic field fluctuations in the Near-LE region appear quite similar to those in the pre-existing solar wind (consistent with the PVI results). The kurtosis analysis revealed further details as more peaked Near-LE distributions for medium and slow sheaths. We also note that in the intertial range spectral indices were the most similar between the solar wind ahead and the Near-LE region. This could be understood considering that the part of the sheath closest to the ejecta leading edge is not affected by the shock and for the
slower sheaths processes at the leading edge do not affect that much either. The Near-LE region of the fast sheath had however distinctly flat $\delta B/B$ distributions and extended tails when compared to the solar wind ahead and the slower sheaths. This could indicate effective magnetic field draping process about the CME ejecta. As discussed by Gosling and McComas (1987) and McComas et al. (1988), the amount of draping increases with the CME speed. The detailed connection of the draping to sheath small-scale structures has however not yet been established. The mean amplitudes and normalised amplitudes of fluctuations
were higher in the sheath than in the preceding solar wind, which is expected as the shock compresses the plasma and field ahead.

The compressibility $\langle \delta |B|/\delta B \rangle$ was also generally higher in the sheath than in the preceding solar wind, suggesting that the new fluctuations that were generated in the transition from the solar wind ahead to the ejecta were mostly compressible. We also found that, in terms of compressibility, sheaths behave more like the slow solar wind than the fast wind, i.e., we found
$\langle \delta |B|/\delta B \rangle$ values above $0.2$ to dominate in the sheath, in agreement with what is found for the slow wind (Bavassano et al., 1982; Bruno and Carbone, 2013). This was the case also for the fastest sheath in our study that was preceded by fast wind, with $\langle \delta |B|/\delta B \rangle$ values below $0.2$.

The inertial range spectral indices in our study for sheath subregions were mostly clearly steeper than Kolmogorov's spectral index of $-1.67$. Borovsky (2012) reported steeper slopes for the magnetic field fluctuations for the slow solar wind (spectral
index $-1.7$) than for the fast wind (spectral index $-1.54$). This also suggests that turbulence in sheaths resembles more turbulence in the slow wind than in the fast wind. We however emphasise that spectral indices in our study exhibited quite large variability. Interestingly also, the fast solar wind ahead the fast sheath had steeper spectral index than the slow solar wind ahead the medium-speed and slow sheaths, which had their spectral indices close to Kraichnan–Iroshinikov's and Kolmogorov's values, respectively. This could be however related to interference of foreshock waves. The kinetic range spectral indices in turn
indicated consistently shallower slopes (spectral indices varying from about $-2.1$ to $-2.5$) than what is on average observed





in the solar wind (spectral index $\sim -2.8$, see Section 3.3). Several previous studies have also reported that in the solar wind both kinetic and inertial spectral indices exhibit large range of values (e.g., Leamon et al., 1998; Smith et al., 2006; Borovsky, 2012). This is the case also in the magnetosheath (e.g., Alexandrova et al., 2008). The inertial range spectral indices in the solar wind tend to also steepen towards Kolmogorov's with increasing distance from the Sun (e.g., Bavassano et al., 1982).

Sahraoui et al. (2009) reported average spectral indices matching our values (from $-2.3$ to $-2.5$) in their study of magnetic field fluctuations in the near-Earth solar wind using Cluster observations for about a 3-h time interval in the frequency range 4–35 Hz (i.e. 2.5–0.03 s). They observed another spectral breakpoint at 35 Hz, close to the electron gyro-scale, after which the slopes further steepened with the spectral index $\sim -3.8$ (see also Sahraoui et al., 2010; Alexandrova et al., 2013; Bruno et al., 2017). Sahraoui et al. (2009) argued that only a relatively small part of the energy is damped at ion scales where kinetic Alfvén

wave cascade still occurs from larger to smaller scales, whereas most of the dissipation occurs in the electron scales where the observed slopes were much steeper. The observations we used in our study are not high-cadence enough to capture this second breakpoint, but the kinetic range spectral indices being close to Sahraoui et al. (2009) implies that this could be generally the case also for CME-driven sheath regions. The statistical study of Huang et al. (2017) found that in the Earth's magnetosheath in the intertial scale the spectral indices were close to $-1$, resembling thus the energy driven scale in the solar wind rather than

Kolmogorov's $-5/3$. Spectral indices close to Kolmogorov's were observed only further away from the bow shock at the flanks and close to the magnetopause. The majority of the events were dominated by compressible magnetosonic-like fluctuations.

We investigated intermittency using various methods. Firstly, we found that the PVI values are generally high in the sheaths when compared to the ambient solar wind, suggesting that sheaths embed frequently intermittent structures. This is in agreement with Zhou et al. (2019). Our analysis of kurtosis and distributions of normalised fluctuations also suggests that sheaths

have in general high intermittency. Generally, intermittency decreases with distance from the Sun and is higher in the fast than in the slow wind, suggesting that it has largely origin at the Sun (Pagel and Balogh, 2001; Wawrzaszek et al., 2015; Bruno, 2019). The slow wind in turn shows strong variability in its intermittency (e.g., Pagel and Balogh, 2002). The fact that CME-driven sheaths feature high intermittency at 1 AU suggests that intermittent structures are actively formed during the formation and evolution of sheaths in interplanetary space. This is consistent with sheaths being heliospheric structures that gather over

long periods of time (up to several days near Earth's orbit) from inhomogeneous solar wind plasma that gets compressed and processed at the CME shock and then piles up at the ejecta leading edge. Good et al. (2020) also found an increase in intermittency for their slow CME sheath from Mercury's to Earth's orbit. The authors suggested that it likely is due to the development of intermittent structures, e.g. current sheets in the sheath during CME propagation.

Finally, we investigated intermittency using the structure function analysis. The standard $p$-model did not match with the

observed $g(m)$ curve features for the majority of the investigated regions. The extended $p$-model however yielded a very good fit both for the Kolmogorov and Kraichnan–Iroshinikov forms and the curves were indistinguishable. This has been the case also in some previous studies in the solar wind (e.g., Horbury and Balogh, 1997; Horbury et al., 1997), and reported also for the studies of turbulence in Earth's magnetosheath (e.g., Yordanova et al., 2008) and in the mangetospheric cusp (e.g., Yordanova et al., 2004). Our results, i.e. that the Kraichnan–Iroshinikov form yielded consistently larger intermittency parameter $p$ and $\alpha$

values, have also been reported earlier (e.g., Tu et al., 1996; Yordanova et al., 2004). In our study the Kraichnan–Iroshinikov





form yielded spectral indices that matched much better to those we obtained from the fitting of the mean magnetic field fluctuations for the corresponding timescales. This suggests that turbulence in the sheath is not fully developed at the orbit of Earth and is more consistent with the Kraichnan–Iroshinikov picture than Kolmogorov's. We note that previous studies have also reported that the Kraichnan–Iroshinikov form produces better fits in particular in cases of solar wind periods with

strong magnetic field (e.g., Podesta, 2011). As discussed above, these findings again reflect the active evolution of sheaths and on-going generation of new fluctuations at the shock and at the ejecta leading edge in particular.

Our study highlights that turbulent properties can vary strongly within the sheath and are controlled by various factors, including the properties of the solar wind ahead, e.g. its plasma beta and level of turbulence, the shock strength and configuration, path of the spacecraft through the shock–sheath–ejecta structure, and the properties of the driving CME ejecta, in particular

by its speed (Kilpua et al., 2013; Moissard et al., 2019). The variations were in general most drastic for the fastest sheath and most subtle for the slowest sheath in our study. We emphasise that we investigated here only one subregion in the middle of the sheath and it would be interesting to study how turbulent properties vary across the whole sheath from the shock to the ejecta leading edge, e.g. by applying a sliding window. Our study also revealed a small-scale coherent structure within the December 14, 2006 fast sheath. This substructure had distinct properties featuring e.g., clearly depressed $\delta B/B$ values,

high compressibility and very shallow spectral slope with intertial range spectral index close to Kraichnan–Iroshinikov's $-3/2$ value. Regarding intermittency, compressibility and general absence of Kolmogorov's type turbulence CME sheaths (in particular in regions close to the shock) are similar to planetary magnetosheath, which implies universality of those properties in a compressible medium, and also the universality regarding the role of the shock on destroying the correlation between the turbulent fluctuations in the solar wind as discussed in Huang et al. (2017). The $f^{-1}$ spectrum was however not found.

## 5   Summary

To summarise, based on our case study of three CME-driven sheath regions observed at the Lagrange L1 point, the magnetic field fluctuation amplitudes, normalised fluctuation amplitudes, and compressibility are enhanced in the sheath when compared to the solar wind ahead. The transition to the sheath thus generates new fluctuations and/or amplifies (in magnitude and/or change in the field direction) of pre-existing fluctuations relative to the mean field. The intermittency was also found to be

higher in the sheath than in the preceding solar wind for many of the investigated subregions. These findings applied for all three sheaths involved, featuring different speeds, shock strengths, and shock angles. Turbulent properties and spectral indices varied also quite considerably between the studied events and the sheath subregions, which reflects the complex structure and formation process of the sheaths of various properties. In general, turbulence in sheaths resembles rather that of the slow solar wind, and we found this to be valid even for the fast sheath preceded by fast wind. According to our study, turbulence in the

sheath is not fully developed, likely due to processes that are constantly in action at the shock and close to the ejecta leading edge as the CME ploughs its way through interplanetary space. This can also explain the high intermittency in sheaths. We also found indications that in sheaths the energy cascade from larger to smaller scales could still be partly on-going at the ion scales, which suggests that the majority of the dissipation would occur at electron scales (not captured by our study).



Future studies focusing deeper on how sheath turbulence varies from the shock to the ejecta leading and extensive statistical studies connecting sheath properties to preceding solar wind and driver characteristics would shed light on these issues. Parker Solar Probe (Fox et al., 2016) and Solar Orbiter (Müller et al., 2013) will also make it possible to analyse sheath properties and turbulence with varying heliospheric distances and possibly provide multi-spacecraft encounters of CME sheaths, allowing to

5    probe how sheath characteristics evolve from the nose of the shock to the CME flanks.

*Competing interests.* The authors declare that they have no conflict of interest.

*Acknowledgements.* he results presented in here have been achieved under the framework of the Finnish Centre of Excellence in Research of Sustainable Space (Academy of Finland grant number 1312390), which we gratefully acknowledge. This project has received funding from the European Research Council (ERC) under the European Union's Horizon 2020 research and innovation programme (ERC-COG 724391).

10    E. K. J. Kilpua acknowledges Academy of Finland project SMASH no. 310445. E. Palmerio's research was supported by the NASA Living With a Star Jack Eddy Postdoctoral Fellowship Program, administered by UCAR's Cooperative Programs for the Advancement of Earth System Science (CPAESS) under award no. NNX16AK22G. E. Yordanova's research was supported by the Swedish Civil Contingencies Agency (MSB, Dnr.2016-2102).



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



**Figure 1.** Interplanetary magnetic field and plasma parameters measured at Wind near L1 for the three analysed events. The panels give the a) magnetic field magnitude, b) magnetic field components in GSE coordinates (blue: $B_X$, green: $B_Y$, red: $B_Z$), c) solar wind speed, d) density, and e) PVI-values calculated for $0.18$, $24$, and $184$ seconds (see text for definition). The orange shaded regions show the 1-hour region in the preceding solar wind and those representing three different parts of the sheath: Near-Shock, Mid-Sheath, and Near-LE regions.

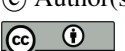

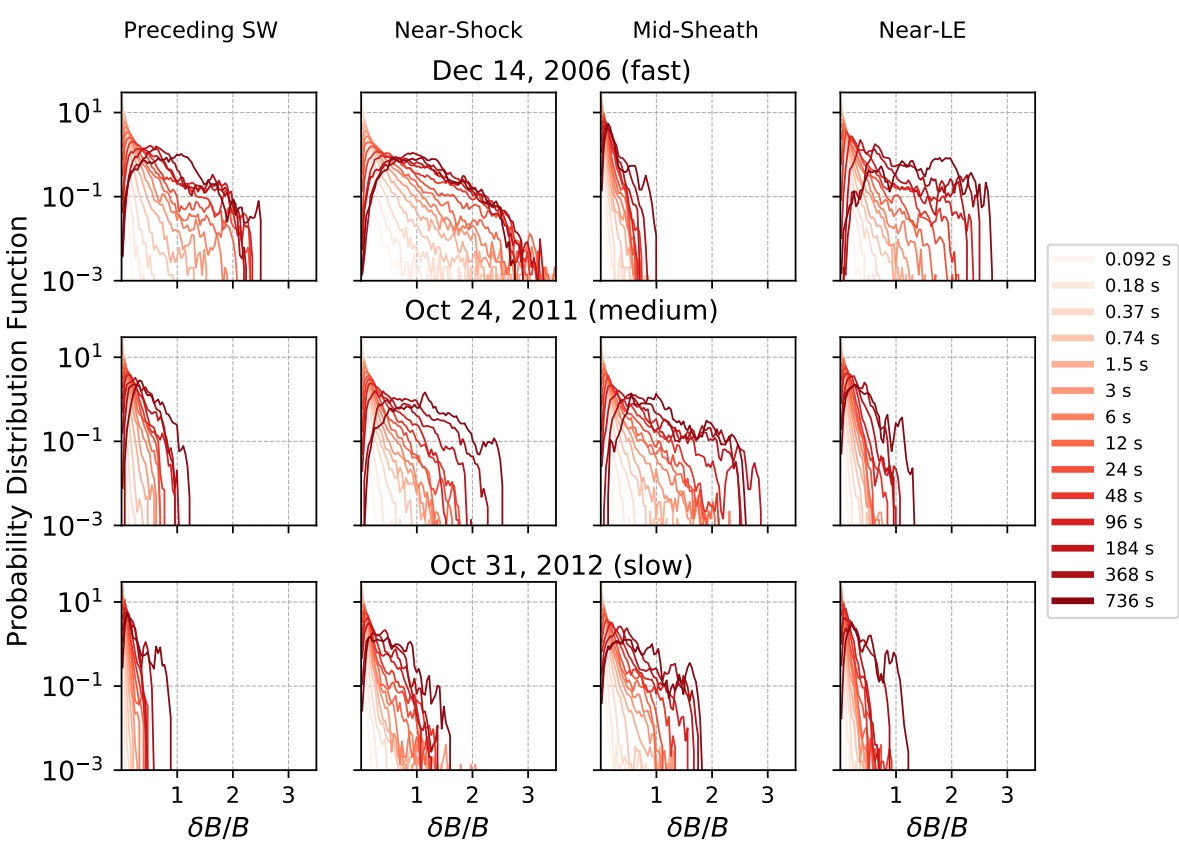

**Figure 2.** Probability distribution functions of $\delta B_Z/B$ for different timescale $\Delta t$ (shown as curves of different colours) for Top) December 14, 2006, Middle) October 24, 2011, and Bottom) October 31, 2012 events. The leftmost columns give the results for the solar wind preceding the shock and the following three columns for three different sheath subregions.





**Figure 3.** Means of fluctuation amplitudes, $\langle \delta B \rangle$, normalized fluctuations, $\langle \delta B/B \rangle$, and fluctuation compressibility, $\langle \delta|B|/B \rangle$), as functions of timescale $\Delta t$. Results are shown separately for the preceding solar wind (crosses) and three different regions of the sheaths (circles). Different colours represent fast (dark blue), intermediate-speed (medium blue), and slow (light blue) sheaths. The dashed vertical lines show the ion cyclotron scales calculated using the magnetic field magnitude averaged over the region in question.

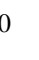



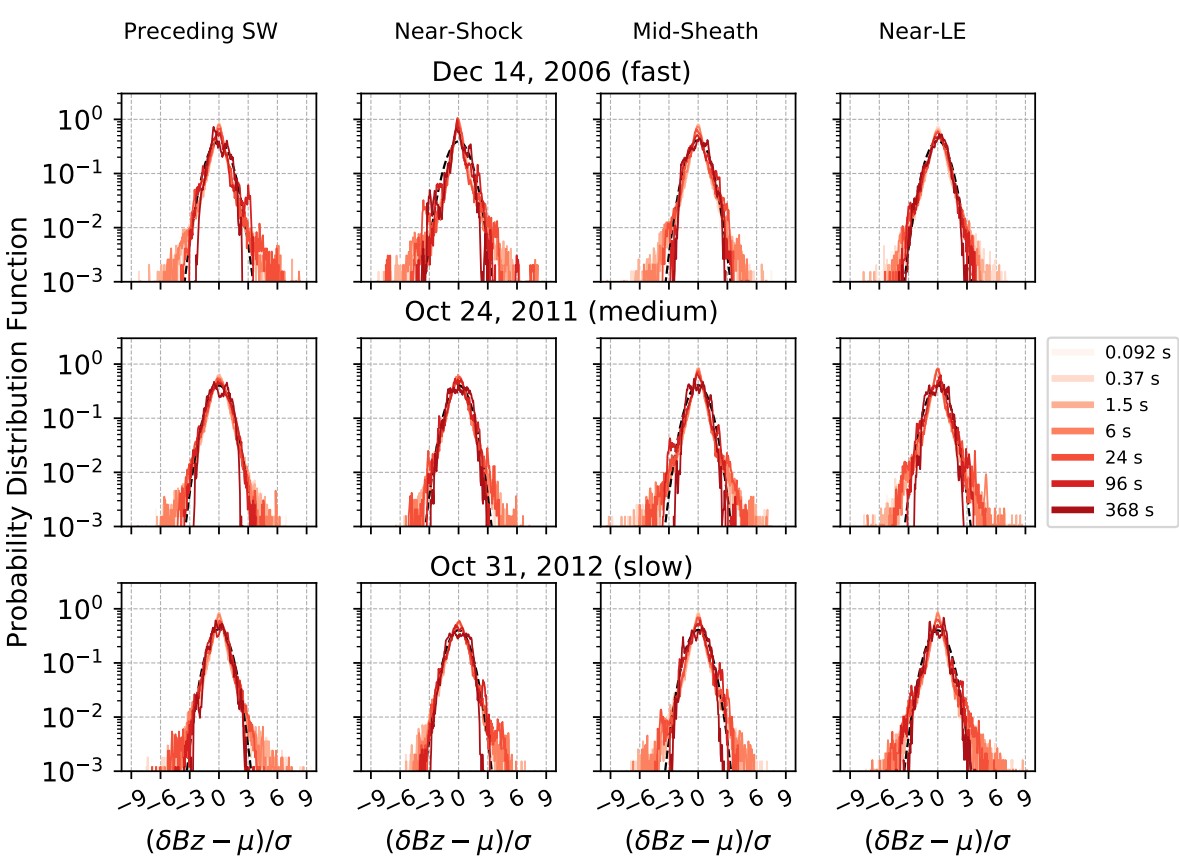

**Figure 4.** Probability distribution functions of $\delta B - \mu/\sigma$ (where $\mu$ is the average and $\sigma$ is the standard deviation of $\delta B_X$ calculated over a 15-minute sliding window) for different timescales $\Delta t$ (shown as curves of different colours) for Top) December 14, 2006, Middle) October 24, 2011, and Bottom) October 31, 2012 events. The leftmost columns give the results for the solar wind preceding the shock and the following three columns for three different sheath subregions. The black dashed line shows the Gaussian for 6 s timescale.

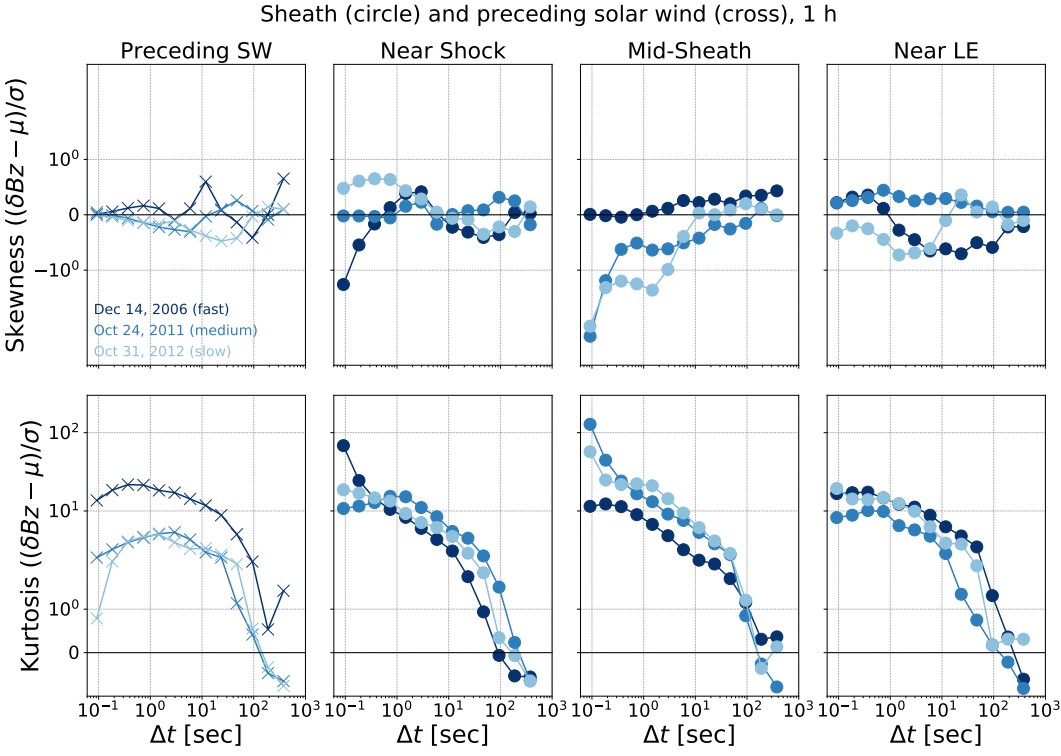

**Figure 5.** Top) Skewness and Bottom) kurtosis calculated for $\delta B$ as a function of timescale $\Delta t$. Different colours and panels represent the fast (dark blue), intermediate-speed (medium blue), and slow (light blue) sheaths. The dots indicate the values in the sheath and the crosses in the preceding solar wind.

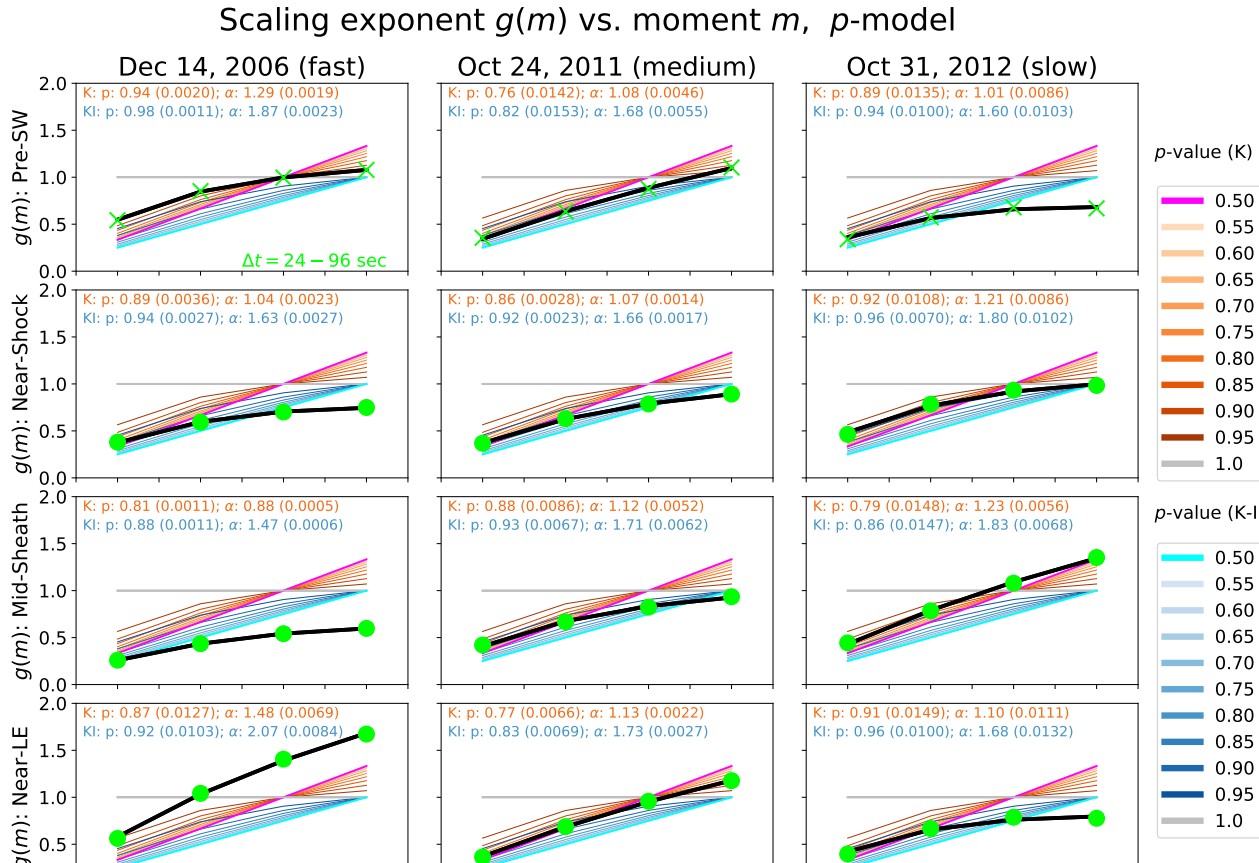

**Figure 6.** The dots and crosses show the scaling exponent $g(m)$ for the structure function $\langle |B(t + \Delta t) - B(t)|^m \rangle$ as a function of moment $m$ ($m$ is set to range from 1 to 4) for the preceding solar wind and three subregions of the sheath. The orange and blue curves show the results for the standard $p$-model with $p$ ranging from 0.5 to 1 in steps of 0.05 for the Kolmogorov's (K) and Kraichnan–Iroshinikov (K–I) forms, respectively. The non-intermittent cases ($p = 0.5$) are shown by pink (K) and cyan (K–I), while the grey horizontal curve gives the maximum intermittency case ($p = 1$). The bright green dots show the results calculated for timescales $\Delta t = 24$–96 s, corresponding to the inertial range. The thick black line shows the least square fits for the extended $p$-model. The values show the values of $p$ and $\alpha$ parameters for the Kolmogorov's (orange) and Kraichnan–Iroshinikov (blue) forms with standard deviation erros in parenthesis.