# Peer review of "Magnetic field fluctuation properties of coronal mass ejection-driven sheath regions in the near-Earth solar wind"

_Annales Geophysicae, 2020_

## Referee Comment (RC1) · Anonymous Referee #1 · 20 May 2020

**General comment**

This manuscript concerns with the investigation of magnetic field fluctuations in three coronal mass ejection (CME)-driven sheath regions at 1 AU with their speeds ranging from slow to fast. The main findings are related to the intermittent and turbulent properties of sheath regions which are also compared and described by means of a common intermittent model as the $p-$model. The authors suggest that turbulent properties in sheaths resemble that of the slow solar wind, that they are partly similar to those found in terrestrial magnetosheath, and that they can vary considerably within the sheath.

[Figure]

In my opinion the results look very interesting and support the view of the complex formation of sheaths and their role in generating fluctuations. The manuscript reads well, contains new results useful for a wide community, and its focus is within the scope of Annales Geophysicae. I have some concerns regarding the presentation of the results and their possible improvements.

**Remarks**

1. Fig. 2 and Fig. 4: why to show both figures instead of only considering Fig. 4? I think it contains much more information than Fig. 2 since it can be easily used to investigate a lot of turbulent properties (as usual) as the (non-)Gaussian behavior. Moreover, I would suggest to go further into the description of non-self-similar properties evident in Fig. 4 for the benefit of the reader.

2. Page 14, lines 19-20: the authors say that "the Kraichnan-Iroshinikov form fits yield consistently larger p-values than the Kolmogorov form fits, but both indicate high intermittency". I suggest the authors to carefully consider some implications of this statement. Both theories are based on similar assumptions although scaling-law behaviors are obtained from HD and MHD equations. So, how to reconcile both theories? I mean the sheaths should be described as a fluid or magnetofluid system? This could affect the larger $p-$values the authors obtained. Moreover, how to assess the suitability of the $p-$model for modeling scaling exponents? From Fig. 6 it seems that some cases are not exactly reproduced through a $p-$model (for example the event on 14 Dec 2006). I would suggest to add a more detailed discussion on these aspects, on possible improvements to simple multifractal models and their suitability in describing sheaths scalings. Furthermore, what about exploring the behavior of singularities and of singularity spectrum derived from scaling exponents? This could also give more information about symmetries and/or irregular/regular behavior of the fractal nature of sheaths. Finally, for the simple benefit of the reader I suggest to only show the best fits through $p-$models in Fig. 6 instead of several $p-$values.

**Minor remarks**

- I suggest to carefully check through the text some inconsistency between text and figures' captions in terms of the magnetic field component used in the analysis. If I correctly understood the authors show only results for $B_z$, while through the text and in figures' captions there are some discrepancies.

- Page 7, lines 20-23: is it possible to measure the degree of compressibility/Alfvénicity?

- Page 7, line 28: it seems to me that it is not a universal property but it depends on the scale. Could the authors comment on this?

- Page 8, line 11: to see how Gaussian are distributions I suggest to also normalize the pdfs with respect to the standard deviations.

- Page 9, line 20: why only three timescales to determine spectral indices?

- Table 2: I suggest to add errors on spectral indices.

- Page 11, Line 21: please check the consistency between components used for the analysis.

- Figure 3: please correct "intertial" with "inertial". Moreover, I suggest to enlarge the frequency range of dashed-line fits to cover more decades.

- Figure 6: please correct the form of the values of $p$ and $\alpha$ parameters. The same number of decimal places should be used for the values and the standard deviations. Moreover, it could be useful for the reader to directly compare the different phases (pre-SW, near-shock, mid-sheath, and near-LE) in the same panel to highlight the different level of intermittency.

---

## Referee Comment (RC2) · Anonymous Referee #2 · 12 Jun 2020

Review of manuscript angeo-2020-17
"Magnetic field fluctuation properties of coronal mass ejection-driven sheath regions …"
by Kilpua et al.

This is a very interesting study and this reviewer enjoyed reading the manuscript. The manuscript should be ready for publication in Annales Geophysicae after minor revision. Below are the specific comments.

1. Page 3, line 11: Riazantseve et al. (2019) used WIND MFI magnetometer data: the SPEKTR-R spacecraft does not yield magnetic-field data.

2. Page 7, line 15: Please add to the manuscript the formula for how $\delta B/B$ is obtained in the time-series data.

3. Page 7, line 21: The reviewer does not understand the statement ", consistent with the absence of strongly compressive fluctuations." Please elaborate on this in the manuscript.

4. Page 8, line 15: What is "the ion cyclotron timescale" and how is it calculated? Isn't the relevant timescale in the spacecraft frame the Doppler-shifted ion gyroradius?

5. Table 2, and elsewhere: Using only 1 hour of data, can you estimate the statistical uncertainty in you value of the inertial-range spectral index? For instance, if you compare your answer to the answer you get for the adjacent hour of data, how big is the variation? Can you comment in the manuscript about whether or not the method you use to obtain the spectral index has less statistical noise than fitting a power spectral density does.

6. Page 1, line 19; Page3, line 20; Page 7, line 10; Page 15, line 9: The shock was quasi-parallel where the WIND spacecraft crossed it, but the magnetic-field direction in the plasma varies significantly and the shock is quasi-parallel and quasi-perpendicular depending on where it is crossed. Looking at the upstream Alfvenic solar wind for the December 14 2006 shock, the magnetic-field varies in time by about $50^\circ$. Hence, the downstream plasma has been shocked in quasi-parallel and quasi-perp fashions and you might not blame its properties on the fact that WIND saw a quasi-parallel shock when it crossed it.

Typos etc.

Page 2, line 32: The sentence "Intermittency in the sheath turbulence grew between the spacecraft." does not make sense.

Page 4, line 11: The phrase "The Near Shock region extends from the shock to the sheath," does not make sense: should "to" be "into"?

Table 1, top row: "Oct 24, 2001 should be "Oct 24, 2011".

Page 9, line 9: ">" should be "<".

Page 14, line 2: $B_z$ was not discussed in the "Introduction".

---

## Author Comment (AC1) · 25 Jun 2020

**General comment**

This manuscript concerns with the investigation of magnetic field fluctuations in three coronal mass ejection (CME)-driven sheath regions at 1 AU with their speeds ranging from slow to fast. The main findings are related to the intermittent and turbulent properties of sheath regions which are also compared and described by means of a common intermittent model as the p−model. The authors suggest that turbulent properties in sheaths resemble that of the slow solar wind, that they are partly similar to those found in terrestrial magnetosheath, and that they can vary considerably within the sheath. In my opinion the results look very interesting and support the view of the complex formation of sheaths and their role in generating fluctuations. The manuscript reads well, contains new results useful for a wide community, and its focus is within the scope of Annales Geophysicae. I have some concerns regarding the presentation of the results and their possible improvements.

**We thank the referee for the detailed reading of our manuscript and constructive comments. We have revised the paper accordingly. Our detailed responses are found from below.**

**Remarks**

1. Fig. 2 and Fig. 4: why to show both figures instead of only considering Fig. 4? I think it contains much more information than Fig. 2 since it can be easily used to investigate a lot of turbulent properties (as usual) as the (non-)Gaussian behavior. Moreover, I would suggest to go further into the description of non-selfsimilar properties evident in Fig. 4 for the benefit of the reader.

**Figure 2 gives important information regarding normalized solar wind fluctuations that are not visible from the other plots in the paper. For example, allowing to see fluctuations exceeding sqrt(2) and 2, signifying fluctuations with significant rotations (over 90 degrees) and those that are at least partly compressional. This type of plots have also been included in some recent solar wind fluctuation studies to which we compare our results in the text (e.g., Chen et al., 2015; Matteini et al., 2018; Good et al., 2020). We now emphasise this in the text and to clarify we have gathered explanations to the beginning of Section 3.2 where Figure 2 is first discussed.**

2. Page 14, lines 19-20: the authors say that "the Kraichnan-Iroshinikov form fits yield consistently larger p-values than the Kolmogorov form fits, but both indicate high intermittency". I suggest the authors to carefully consider some implications of this statement. Both theories are based on similar assumptions although scaling-law behaviors are obtained from HD and MHD equations. So, how to reconcile both theories? I mean the sheaths should be described as a fluid or magnetofluid system? This could affect the larger p−values the authors obtained. Moreover, how to assess the suitability of the p−model for modeling scaling exponents? From Fig. 6 it seems that some cases are not exactly reproduced through a p−model (for example the event on 14 Dec 2006). I would suggest to add a more detailed discussion on these aspects, on possible improvements to simple multifractal models and their suitability in describing sheaths scalings. Furthermore, what about exploring the behavior of singularities and of singularity

spectrum derived from scaling exponents? This could also give more information about symmetries and/or irregular/regular behavior of the fractal nature of sheaths. Finally, for the simple benefit of the reader I suggest to only show the
best fits through p−models in Fig. 6 instead of several p−values

**These are all very relevant points. It indeed appears that the used models give partly contradictory results and it is reasonable to question to what extent they can be used to describe turbulence in CME driven sheaths. We have also now replaced the p-model figure to show the results for the sum of the structure function(similar to Pei et al., 2016, added as a reference) instead of the Bz component. The results are now calculated over five timescales in the inertial range (6 to 96 seconds). We thank the reviewer for interesting suggestions for a deeper analysis, but would like to save them for future work. We have however extended the discussion and do not make so strong conclusions based on the p-model. The analysis indeed hints that these models may not be well suited to describe turbulence in CME driven sheaths. Also we now cite Pagel et al.,2002 that deviations from g(3)=1 and g(4)=1 can give rise to false non-intermittent signatures. Agreement with Kraichnan form would suggest that sheaths behave more like magnetofluid, we mention this in the text now. This is consistent with sheaths having generally clearly enhanced magnetic field magnitudes. We also refer now to Li et al., 2012 who argued that in solar wind spectra would steepen from Kraichnan-like to more Kolmogorov-like spectra due to presence of intermittent current sheets.**

**We would like to keep the colored lines in the Figure 6 as they show how observed (lime) points do not fit to standard p-model curves (i.e. do not follow their shapes in most cases). We have however made these lines slightly fainter so that they are more in the background.**

**Minor remarks**

suggest to carefully check through the text some inconsistency between text and figures' captions in terms of the magnetic field component used in the analysis. If I correctly understood the authors show only results for Bz, while through the text and in figures' captions there are some discrepancies.

**Corrected**

Page 7, lines 20-23: is it possible to measure the degree of compressibility/Alfvénicity?

**Figure 3 gives information about compressibility. In addition, this information comes also from Figure 2 where large dB/B values (> 2) suggest that fluctuation must be at least partly non-compressible (i.e. non-Alfvénic)**

Page 7, line 28: it seems to me that it is not a universal property but it depends on the scale. Could the authors comment on this?

**We are not sure where the reviewer refers to this comment.**

Page 8, line 11: to see how Gaussian are distributions I suggest to also normalize the pdfs with respect to the standard deviations.

**As this info shown more clearly in Figure 4 we now refer to Section 3.5.1 at this point (and discussion there).**

Page 9, line 20: why only three timescales to determine spectral indices?

**For the kinetic range we cannot include more timescales due to Wind time resolution available. However, we have now included five time scales for the inertial range. This resulted in some changes in the indices, but the main conclusions of our study remain.**

Table 2: I suggest to add errors on spectral indices.

**We have added standard errors of deviations from the fitting in Table 2 in parenthesis.**

Page 11, Line 21: please check the consistency between components used for the analysis.

**We have added PDFs for Bx and By components to Supplementary Materials. The PDFs look overall very similar, but the Bz has somewhat more extended tails than Bx/By. This is also visible from Figure 1 as larger amplitude changes in Bz than in the other components. This could imply slightly larger intermittency for the Bz component. We have also now replaced the structure function for Bz shown in Figure 6 with the sum of structure functions for each component.**

Figure 3: please correct "intertial" with "inertial". Moreover, I suggest to enlarge the frequency range of dashed-line fits to cover more decades.

**We have corrected "inertial" in the figure and also a few similar typos elsewhere in the paper. We also now include 5 timescales to calculate the slopes in the inertial range and accordingly show the dashed-line to cover these extended decades.**

Figure 6: please correct the form of the values of p and  parameters. The same number of decimal places should be used for the values and the standard deviations. Moreover, it could be useful for the reader to directly compare the different phases (pre-SW, near-shock, mid-sheath, and near-LE) in the same panel to highlight the different level of intermittency

**We have corrected these. We think the standard p-model curves also allow the reader to sufficiently compare between the events.**

---

## Author Comment (AC2) · 25 Jun 2020

This is a very interesting study and this reviewer enjoyed reading the manuscript. The manuscript should be ready for publication in Annales Geophysicae after minor revision. Below are the specific comments.

**We thank the referee for the detailed reading of our manuscript and constructive comments. We have revised the paper accordingly. Our detailed responses are found from below.**

1. Page 3, line 11: Riazantseve et al. (2019) used WIND MFI magnetometer data: the SPEKTRR spacecraft does not yield magnetic-field data.

**We have revised the text accordingly. Thank you for pointing this out.**

2. Page 7, line 15: Please add to the manuscript the formula for how δB/B is obtained in the time-series data.

**We have defined this now in the text.**

3. Page 7, line 21: The reviewer does not understand the statement ", consistent with the absence of strongly compressive fluctuations." Please elaborate on this in the manuscript.

**We have modified this part and explain this now in more detail in the beginning of the subsection.**

4. Page 8, line 15: What is "the ion cyclotron timescale" and how is it calculated? Isn't the relevant timescale in the spacecraft frame the Doppler-shifted ion gyroradius?

**We have detailed this in the text.**

5. Table 2, and elsewhere: Using only 1 hour of data, can you estimate the statistical uncertainty in you value of the inertial-range spectral index? For instance, if you compare your answer to the answer you get for the adjacent hour of data, how big is the variation? Can you comment in the manuscript about whether or not the method you use to obtain the spectral index has less statistical noise than fitting a power spectral density does.

**The use of 1-hour region was based on selecting regions of CME sheaths where fluctuations are generated primarily by certain physical mechanisms (e.g., near leading edge by processes associated with draping and expansion of the ejecta). We have however added now the errors from the fitting procedure to the table. We are not aware of exact comparison of noise compared to fitting to PSD but would expect it to be less. We have explained this part now in more detail and refer to Matteini et al., 2018 and Good et al., 2020 as examples of studies where this approach has been applied before.**

---

## Author Response (AR1)

The comment was uploaded in the form of a supplement:
https://angeo.copernicus.org/preprints/angeo-2020-17/angeo-2020-17-AC1-supplement.pdf